# Autocrine Regulation of Interleukin-6 via the Activation of STAT3 and Akt in Cardiac Myxoma Cells

**DOI:** 10.3390/ijms25042232

**Published:** 2024-02-13

**Authors:** Michihisa Jougasaki, Yoko Takenoshita, Katsuyuki Umebashi, Masayoshi Yamamoto, Ku Sudou, Hitoshi Nakashima, Masahiro Sonoda, Tamahiro Kinjo

**Affiliations:** Institute for Clinical Research, NHO Kagoshima Medical Center, Kagoshima 892-0853, Japan; takenoshita.yoko.cj@mail.hosp.go.jp (Y.T.); umebashi.katsuyuki.gc@mail.hosp.go.jp (K.U.); yamamoto.masayoshi.bc@mail.hosp.go.jp (M.Y.); pd3z8uos@okayama-u.ac.jp (K.S.); nakashima.hitoshi.mc@mail.hosp.go.jp (H.N.); sonoda.masahiro.by@mail.hosp.go.jp (M.S.); kinjo.tamahiro.tr@mail.hosp.go.jp (T.K.)

**Keywords:** myxoma, interleukin-6, autocrine, STAT3, Akt

## Abstract

Plasma concentrations of a pleiotropic cytokine, interleukin (IL)-6, are increased in patients with cardiac myxoma. We investigated the regulation of IL-6 in cardiac myxoma. Immunohistochemical staining and reverse transcription-polymerase chain reaction (RT-PCR) revealed that IL-6 and its receptors, IL-6 receptor (IL-6R) and gp130, co-existed in the myxoma cells. Myxoma cells were cultured, and an antibody array assay showed that a conditioned medium derived from the cultured myxoma cells contained increased amounts of IL-6. Signal transducer and activator of transcription (STAT) 3 and Akt were constitutively phosphorylated in the myxoma cells. An enzyme-linked immunosorbent assay (ELISA) showed that the myxoma cells spontaneously secreted IL-6 into the culture medium. Real-time PCR revealed that stimulation with IL-6 + soluble IL-6R (sIL6R) significantly increased IL-6 mRNA in the myxoma cells. Pharmacological inhibitors of STAT3 and Akt inhibited the IL-6 + sIL-6R-induced gene expression of IL-6 and the spontaneous secretion of IL-6. In addition, IL-6 + sIL-6R-induced translocation of phosphorylated STAT3 to the nucleus was also blocked by STAT3 inhibitors. This study has demonstrated that IL-6 increases its own production via STAT3 and Akt pathways in cardiac myxoma cells. Autocrine regulation of IL-6 may play an important role in the pathophysiology of patients with cardiac myxoma.

## 1. Introduction

Interleukin (IL)-6 is a multifunctional cytokine that plays important roles in immune regulation, inflammation, metabolism, and tissue regeneration [1,2]. In IL-6 classic signaling, IL-6 binds to IL-6-transmembrane receptor (IL-6R) and subsequent recruitment of the transmembrane gp130, activating the intracellular signal transduction system, including Janus kinase (JAK)-signal transducer and activator of transcription (STAT) 3, phosphatidylinositol-3-kinase (PI3K)/Akt, and mitogen-activated protein kinase [3]. A soluble form of IL-6R (sIL-6R) is released from the cell surface by proteolysis or by IL-6 mRNA alternative splicing. The cells lacking transmembrane IL-6R are not responsive to IL-6; however, they respond to IL-6 in the presence of sIL-6R. Thus, the IL-6 + sIL-6R complex binds to transmembrane gp130 on the cells that do not express transmembrane IL-6R, and this process is known as IL-6 trans-signaling [4]. Nowadays, sIL-6R is considered as an agonist molecule, allowing IL-6 to have an effect on the cells lacking IL-6R but ubiquitously expressing transmembrane gp130 [5]. In addition, IL-6 is produced by a variety of cells, such as T cells, B cells, monocytes-macrophages, fibroblasts, epidermal keratinocytes, endothelial cells, mesangial cells, and tumor cells, including cardiac myxoma cells in the human body [6].

Cardiac myxoma is the most common primary tumor of the heart [7]. The clinical presentation of cardiac myxoma typically involves triad of intracardiac obstruction, embolization, and constitutional symptoms [8]. Accumulating evidence has revealed that cardiac myxomas spontaneously produce IL-6 [9,10,11,12,13,14,15,16,17], which is associated with the constitutional symptoms in patients with cardiac myxoma. Immunohistochemical studies in patients with cardiac myxoma demonstrated that 70 to 80% of tumors showed immunohistochemical expression of IL-6 [18,19]. Increased gene expression of IL-6 was also found in the cardiac myxoma tissue by using the method of reverse transcription-polymerase chain reaction (RT-PCR) [17] and in situ hybridization [13]. A positive correlation between plasma levels of IL-6 and tumor size was observed in patients with cardiac myxoma [16,20,21]. Specifically, Endo et al. showed that patients with constitutional symptoms had significantly larger tumors than those without constitutional signs, suggesting that the production of IL-6 increased in proportion to tumor size [22]. In contrast, other investigators reported that peripheral monocytes contributed to the elevated production of IL-6 in patients with cardiac myxoma [23]. The regulation of IL-6 in patients with cardiac myxoma has not been clarified yet.

In the present study, we hypothesized that IL-6 was produced in an autocrine and paracrine manner through the activation of STAT3 and Akt signaling pathways in the cardiac myxoma cells. Therefore, this study was designed to investigate the production and secretion of IL-6 in the cardiac myxoma cells, specifically focusing on the signal transduction system of STAT3 and Akt pathways.

## 2. Results

### 2.1. Patient Profile

A 70-year-old woman presented with transient dizziness due to ischemic cerebral infarctions. Laboratory tests showed a leukocyte count of 9370 cells/mL, C-reactive protein of 0.48 mg/dL, and IL-6 of 3.29 pg/mL. Echocardiographic examination in search of possible embolic sources for cerebral infarction showed a large mass in the left atrium. Cardiac surgery was performed and ellipsoidal 50 × 80 × 55 mm^3^ smooth and glossy tumor adherent to the interatrial septum was resected from the heart. Histological examination revealed that the polygonal, spindle-shaped or stellate cells which had oval nuclei and eosinophilic cytoplasm, exhibiting cords or nests, were surrounded by mucoid matrix (HE staining in Figure 1). The nuclei of the tumor cells were without pleomorphism or mitosis, and the tumor was diagnosed as cardiac myxoma. The study protocol was approved by the institutional review board, and informed consent was given by the patient.

### 2.2. Immunohistochemical Staining in the Cardiac Myxoma Tissue

Immunohistochemical staining revealed positive immunoreactivities for IL-6, IL-6R, and gp130 in the polygonal, spindle-shaped, or stellate cells of cardiac myxoma (Figure 1). In addition, these myxoma cells showed a strong expression of calretinin. No immunoreactivities were detected in the myxoma tissue when treated with normal IgG instead of the respective primary antibodies.

### 2.3. Immunocytochemical Staining in the Cardiac Myxoma Cells

Immunocytochemical staining showed positive immunoreactivities for IL-6, IL-6R, and gp130 in the cultured myxoma cells (Figure 2). Immunoreactivities for calretinin was also positive in the myxoma cells. The cells treated with normal IgG instead of the respective primary antibodies demonstrated no immunoreactivities.

### 2.4. Gene Expression of IL-6 and Its Receptor Complexes in the Cardiac Myxoma Cells

To investigate the gene expressions of IL-6, and its receptor complexes, IL-6R and gp130 in the cardiac myxoma cells, total RNA was extracted from the cultured myxoma cells, examined by using RT-PCR, and the results were compared with those in the cultured human umbilical vein endothelial cells (HUVECs). RT-PCR with specific primers demonstrated that both myxoma cells and HUVECs yielded positive products for IL-6, and its receptor complexes, IL-6R and gp130 (Figure 3). However, gene expression of calretinin was detected only in the myxoma cells, but not in the cultured HUVECs.

### 2.5. Antibody Array Assay

The antibody array assay was used to investigate the protein secreted from the cultured myxoma cells. The culture supernatant of myxoma cells incubated for 7 days were analyzed using a dot blot array. IL-6 was markedly increased in the supernatant of the myxoma cells compared with control culture media (Figure 4). In addition to IL-6, monocyte chemoattractant protein (MCP)-1, granulocyte-macrophage colony-stimulating factor (GM-CSF), and platelet-derived growth factor (PDGF)-BB were also increased in the supernatant of the cultured myxoma cells.

### 2.6. Culture Supernatant of the Myxoma Cells Induced STAT3 Phosphorylation in HUVECs

We next examined whether the culture supernatant derived from the myxoma cells have transferrable biologic properties to stimulate STAT3 phosphorylation in cultured HUVECs. The myxoma cells were incubated for 7 days, and the supernatant was added to HUVECs, incubated for 30 min, and subjected to Western immunoblot analysis. Culture supernatant derived from the myxoma cells caused activation of STAT3, which was attenuated by the addition of anti-IL-6 antibody, but not by control IgG (Figure 5), suggesting that STAT3 activating properties in the culture supernatant were due, at least in part, to the biological activity of transferable IL-6.

### 2.7. Constitutive Phosphorylation of STAT3 and Akt in the Cardiac Myxoma Cells

To determine the phosphorylation status of STAT3 and Akt, the cultured myxoma cells were washed with phosphate-buffered saline, incubated in fresh medium, and left untreated for 24 h. As shown in Figure 6, Western immunoblot analysis demonstrated that STAT3 and Akt were constitutively phosphorylated according to time within 24 h. STAT3 was gradually phosphorylated from 16 to 24 h, although total STAT3 protein was unchanged for 24 h. Phosphorylation of Akt occurred from 4 to 24 h, which was earlier than STAT3 phosphorylation. Similarly, the total Akt protein was not changed for 24 h. On the other hand, STAT1 was not phosphorylated for 24 h.

### 2.8. Secretion of IL-6 from the Cardiac Myxoma Cells

To further examine whether cultured myxoma cells actually secrete IL-6 to the culture medium, the cells were washed with phosphate-buffered saline and incubated in the fresh medium. The supernatant was collected at 8, 16, and 24 h, and the concentration of IL-6 was measured by ELISA. The results were compared with those of HUVECs. Cultured myxoma cells spontaneously secreted considerable amounts of IL-6 into the culture medium from 8 to 24 h (Figure 7A). The addition of antibody against IL-6R or gp130 reduced the secretion of IL-6 from the myxoma cells (Figure 7B). To elucidate the signal transduction pathways accounting for the secretion of IL-6, pharmacological inhibitors were used to interfere with STAT3 and Akt pathways. As shown in Figure 7C, the secretion of IL-6 from the myxoma cells was significantly inhibited by the treatment with AG490 (JAK2 inhibitor, 100 μmol/L), piceatannol (STAT1/3 inhibitor, 100 μmol/L), LLL12 (STAT3 inhibitor, 10 μmol/L), and Ly294002 (PI3K/Akt inhibitor, 30 μmol/L) (Figure 7C). 

### 2.9. Effects of STAT3 siRNA Transfection on the Secretion of IL-6 from the Cardiac Myxoma Cells

To further confirm the role of STAT3 in the secretion of IL-6, we transfected antibody-free myxoma cells with STAT3 siRNA. The efficacy of the transfection of siRNA is shown in Figure 8A. Transfection of STAT3 for 48 h reduced STAT3 expression by 56%. Transfection of STAT3 significantly attenuated spontaneous IL-6 secretion from the myxoma cells compared with control-scrambled siRNA transfection (Figure 8B).

### 2.10. Effects of IL-6 + sIL-6R on the Gene Expression of IL-6 in the Cardiac Myxoma Cells

Real-time PCR demonstrated that treatment with IL-6 + sIL-6R resulted in an increase in IL-6 mRNA in a time-dependent manner with a statistical significance at 4 to 24 h (Figure 9A). To examine whether signaling pathways, such as STAT3 and Akt, were involved in the IL-6 + sIL-6R-induced IL-6 autoregulation, the cultured myxoma cells were pre-treated with pharmacological inhibitors of STAT3 and Akt for 2 h, followed by stimulation with IL-6 + sIL-6R for 4 h to measure IL-6 mRNA in the myxoma cells. Real-time PCR demonstrated that IL-6 + sIL-6R-induced increase in IL-6 mRNA was significantly suppressed by the pretreatment with JAK-STAT inhibitors, such as AG490, piceatannol, and LLL12 in the myxoma cells (Figure 9B). Akt inhibitor, Ly294002 also decreased IL-6 + sIL-6R-induced increase in IL-6 mRNA as well.

### 2.11. Effects of IL-6 + sIL-6R on the Phosphorylation of STAT3 and Akt in the Cardiac Myxoma Cells

The cultured myxoma cells were exposed to IL-6 + sIL-6R for different time periods (5 to 120 min), and their protein extracts were examined by Western immunoblot analysis. IL-6 + sIL-6R stimulated the phosphorylation of STAT3, peaking at 15 to 30 min, and declined thereafter (Figure 10A). On the other hand, IL-6 + sIL-6R induced phosphorylation of Akt at 5 to 15 min and declined at 30 min (Figure 10B).

### 2.12. Immunofluorescence Staining

To examine whether inhibition of JAK-STAT pathway affects the translocation of phosphorylated STAT3 to the nucleus, the cultured myxoma cells were pretreated with AG490 followed by incubation with IL-6 + sIL-6R for 30 min. The immunofluorescence signal of phosphorylated STAT3 located in the nuclei of the myxoma cells after incubation with IL-6 + sIL-6R for 30 min was compared with the untreated control cells (Figure 11). The translocation of phosphorylated STAT3 to the nucleus induced by IL-6 + sIL-6R was inhibited by AG490 at the dose of 100 μmol/L.

## 3. Discussion

The present study demonstrated that human cardiac myxoma cells express IL-6 and its receptor complexes, IL-6R and gp130. Immunohistochemical staining in the excised cardiac myxoma tissue as well as immunocytochemical staining in the cultured cardiac myxoma cells demonstrated that IL-6 and its receptor complexes, such as IL-6R and gp130, were co-localized in the cardiac myxoma cells. In addition, the RT-PCR study showed the expressions of IL-6 mRNA, IL-6R mRNA, and gp130 mRNA in the myxoma cells, together with calretinin mRNA. Calretinin is a calcium-binding protein principally expressed in neurons [24]. Nowadays, calretinin is considered as a reliable marker for cardiac myxoma [25,26,27], and, therefore, we used the protein and gene expression of calretinin as a validation analysis for the myxoma cells. The presence of ligand and receptors of IL-6 in the myxoma cells supports the idea that both the classic and trans-signaling pathway are involved in the signal transduction system of IL-6 in the human cardiac myxoma cells. It is difficult to measure IL-6 protein levels in the presence of exogenously added IL-6. Therefore, we analyzed the gene expression of IL-6 mRNA to investigate the role of IL-6 on the myxoma cells. In the present study, myxoma cells were treated with exogenous IL-6 + sIL-6R, and mRNA was isolated and analyzed by real-time PCR. Thus, we analyzed trans-signaling by using IL-6 + sIL-6R for activation of IL-6 mRNA and found that IL-6 trans-signaling caused autocrine activation of IL-6 gene expression through STAT3 and PI3K/Akt pathways in the myxoma cells.

The clinical characteristics of cardiac myxoma include intracardiac obstruction, embolization, and constitutional symptoms such as malaise, fever, anorexia, arthralgia, and weight loss [8]. Since the initial study by Hirano et al. [9], accumulating evidence has demonstrated that cardiac myxomas constitutively produce considerable amounts of IL-6 [10,11,12,13,14,15,16,17], explaining the underlying pathophysiology of inflammatory and immune features of constitutional symptoms observed in these patients with cardiac myxoma. Previous studies reported that plasma levels of IL-6 were positively correlated with tumor size in patients with cardiac myxoma [16,20,21,22,28]. Mendoza et al., showing the significant correlation between serum IL-6 concentration and primary tumor size, has demonstrated that the overproduction of IL-6 by cardiac myxoma is responsible for the immunological abnormalities and constitutional symptoms seen in patients with cardiac myxoma [20]. Endo et al. analyzed constitutional signs in 204 patients with cardiac myxoma, in whom 90 cases were with constitutional signs, 92 without constitutional signs, with 22 classified as undetermined [22]. The authors demonstrated that patients with constitutional signs had significantly larger tumors than those without constitutional signs, suggesting that the production of IL-6 increases in proportion to tumor size, and that constitutional signs result when IL-6 concentrations exceed a certain threshold [22]. More precise analyses of the relationship between constitutional symptoms and plasma IL-6 concentrations are warranted.

In the present study, the cardiac myxoma cells constitutively produced and secreted considerable amounts of IL-6. IL-6 concentrations in the supernatant of myxoma cells increased in a time-dependent manner. As for the source of IL-6 in patients with cardiac myxoma, the present cell culture study, together with other previous studies [9,10,14,29,30], directly demonstrated that IL-6 was produced and secreted from the cardiac myxoma cells in vitro. Therefore, it seems likely that circulating IL-6 in patients with cardiac myxoma is derived from the neoplastic myxoma cells. On the other hand, Garcia-Zubiri et al. demonstrated the contribution of peripheral monocytes to the increased IL-6 serum levels in a patient with cardiac myxoma [23]. The authors presented a 69-year-old patient with cardiac myxoma, in which 74.4% of peripheral blood monocytes produced IL-6, and the percentage of monocytes producing IL-6 was significantly decreased at one month after surgical resection of the cardiac myxoma [23]. Morishima et al. reported a rare case of IL-6-producing cardiac myxoma resembling multicentric Castleman’s disease with lymphadenopathy and abnormal plasma cell infiltration in bone marrow [31]. In addition, other investigators reported that IL-6 secreted by the cardiac myxoma caused the mediastinal lymphadenopathy, and resection of the myxoma tissue resulted in resolution of mediastinal lymphadenopathy together with a reduction in IL-6 levels to normal levels [32,33]. In the present study, Figure 5 showed that secreted IL-6 in the supernatant was biologically transferable and caused STAT3 activation in vascular endothelial cells that were adjacent to myxoma cells in the body. Therefore, it seems possible that IL-6 produced by the neoplastic myxoma cells provokes the activation of monocytes–macrophages to secrete additional IL-6, causing mediastinal lymphadenopathy and multicentric Castleman’s disease in patients with cardiac myxoma. Likewise, normal cardiac myocytes might also be influenced by IL-6 secreted from the neoplastic cardiac myxoma cells and might cause ventricular hypertrophy in a patient with cardiac myxoma [34]. Further studies are needed to address these issues.

Since the initial study by Hirano et al. [9], several groups of investigators have isolated and cultured the cardiac myxoma cells, and secreted molecules, including IL-6, in the supernatant have been investigated [10,14,29,30,35]. Besides IL-6, Sakamoto et al. reported that the culture supernatant of cardiac myxoma cells contained significant amounts of IL-8, growth-regulated oncogene (GRO)-α, endothelin (ET)-1, big ET-1, and vascular endothelia growth factor (VEGF) [14,29,30,35]. The present antibody array assay analysis demonstrated that, in addition to IL-6, the culture supernatant of myxoma cells incubated for 7 days included considerable amounts of molecules such as MCP-1, GM-CSF, and PDGF-BB. These findings were supported by findings in previous investigations indicating that these bioactive molecules were expressed in the cardiac myxomas [16,36,37]. Zhang et al. demonstrated in an immunohistochemical analysis of 17 cardiac myxoma tissues that MCP-1 was found in the cytoplasm of the myxoma cells, and that the proportions of MCP-1-positive myxoma cells were significantly correlated with an increased micro-vessel count [37], suggesting the role of MCP-1 in angiogenesis associated with tumor growth. We and other investigators reported that IL-6 induced MCP-1 in human vascular endothelial cells [38,39] and peripheral mononuclear cells [40], suggesting the possibility that IL-6 secreted from the cardiac myxoma cells stimulates MCP-1 production in the myxoma cells or various other cells around. On the other hand, stimulation of vascular smooth muscle cells with MCP-1 resulted in a concentration- and time-dependent secretion of IL-6 [41]. In addition, by using immunohistochemical staining procedures, Gaumann et al. showed that PDGF-BB was present in the cardiac myxoma cells [36]. PDGF-BB increased the expression of IL-6 in cultures of osteoblasts from fetal rat calvariae [42]. Soeparwata et al. reported that one patient out of four patients with cardiac myxoma, in whom plasma GM-CSF concentrations were measured, showed an increased level of GM-CSF [16]. IL-6 is a potent inducer of GM-CSF expression by post-transcriptional stabilization of the GM-CSF mRNA [43], and GM-CSF significantly increased Il-6 secretion via extracellular signal-regulated kinase (ERK)1/2 pathway in macrophages [44]. As mentioned above, cardiac myxoma cells produce significant amounts of IL-8, GRO-α, ET-1, and VEGF, in addition to IL-6 in vitro. The present study added MCP-1, PDGF-BB, and GM-CSF as new members to the secreted substances of the cardiac myxoma cells. These biologically active molecules have pleiotropic properties. Although the precise causative and functional roles of these factors in the cardiac myxoma cells remain unknown, IL-6 and these substances have an important role in inflammation, tumor growth, angiogenesis, and tumor cell migration in the cardiac myxomas. Crosstalk among these molecules in the cardiac myxoma cells is intriguing, and further studies are required to elucidate the role of bioactive molecules secreted from the cardiac myxoma cells.

In the present study, stimulation with IL-6 + sIL-6R resulted in an increase in IL-6 mRNA, and pharmacological inhibitors against JAK/STAT3 and PI3L/Akt inhibited the IL-6 + sIL-6R-induced activation of IL-6 mRNA in the myxoma cells. Likewise, Franchimont et al. demonstrated that IL-6 + sIL-6R induced IL-6 mRNA in rat osteoblastic cells at the transcriptional levels, enhancing IL-6 rates of transcription and promoter activity [45]. The present study also demonstrated that STAT3 was constitutively phosphorylated together with spontaneous secretion of IL-6 in the cultured myxoma cells. Constitutive STAT3 phosphorylation, in association with IL-6 secretion, has been demonstrated in previous investigations [46,47,48,49]. Huang et al. showed that the lung adenocarcinoma cells spontaneously secreted IL-6 and possessed constitutively activated STAT3, and that inhibitors of JAK2/STAT3 and PI3-K/Akt pathways downregulated IL-6 secretion in these cells [46]. Schuringa et al. reported that the autocrine and paracrine secretion of IL-6 caused the constitutive activation of STAT3 in acute myelogenous leukemia cells [47]. A small GTPase, Rac1, stimulated STAT3 activation through the induction of an autocrine IL-6 feedback loop that leads to the activation of the JAK/STAT pathway [50]. Hirano reported an amplification mechanism for the production of IL-6 and various other cytokines and chemokines through a synergistic interaction between STAT3 and nuclear factor kappa B (NF-κB) and advocated “IL-6 amplifier (IL-6 Amp)” [51]. IL-6 might exert its biological actions as an autocrine and/or paracrine factor in the cardiac myxoma cells in a similar fashion as other human tumor cells [46,47,48,52]. Further studies are needed for the precise role of the production of IL-6 in patients with cardiac myxoma.

## 4. Materials and Methods

### 4.1. Reagent

Recombinant IL-6 and sIL-6R were purchased from Pepro Tech (Rocky Hill, NJ, USA). The monoclonal antibodies against IL-6, gp130, and β-actin were from Santa Cruz Biotechnology (Heidelberg, Germany). The polyclonal antibodies against STAT1, phospho-STAT1 (Tyr701), STAT3, phospho-STAT3 (Tyr705), Akt, phospho-Akt (Ser473) were obtained from Cell Signaling Technology (Beverly, MA, USA). The polyclonal antibodies against IL-6R and calretinin were purchased from GeneTex (Irvine, CA, USA). AG490 (JAK2 inhibitor), piceatannol (STAT1/3 inhibitor), and LY294002 (PI3K/Akt inhibitor) were purchased from FUJIFILM Wako Pure Chemical (Osaka, Japan). LLL12 was from BioVision (Milpitas, CA, USA). The reagents of siRNA were obtained from Santa Cruz Biotechnology (Heidelberg, Germany).

### 4.2. Culture of Cardiac Myxoma Cells

Cardiac myxoma tissue obtained by surgery was minced into small pieces, and dissected tissues were treated with collagenase and trypsin before being dissociated and plated in plastic plates precoated with type I collagen (Asahi Techno Glass, Nagoya, Japan). The cells were maintained in culture media containing TIL Media I (IBL, Fujioka, Japan)/Medium 199 (Thermo Fisher Scientific, Waltham, MA, USA) (1:1) supplemented with 10% fetal bovine serum (Life Technologies, Carlsbad, CA, USA), 0.5 mg/mL fungizone, 0.25 mg/mL amphotericin B, 100 mg/mL streptomycin, and 100 U/mL penicillin (Life Technologies, Carlsbad, CA, USA). The cells were incubated at 37 °C in a humidified incubator with 5% CO_2_.

### 4.3. Culture of Human Umbilical Vein Endothelial Cells (HUVECs)

HUVECs were purchased from Kurabo (Osaka, Japan), and seeded in plastic plates precoated with type I collagen (Asahi Techno Glass, Nagoya, Japan). The cells were maintained in an endothelial cell growth medium (Promo cell, Heidelberg, Germany) supplemented with 0.5 mg/mL fungizone, 0.25 mg/mL amphotericin B, 100 mg/mL streptomycin, 100 U/mL penicillin (Life Technologies, Carlsbad, CA, USA). The cells were kept at 37 °C in a humidified incubator with 5% CO_2_.

### 4.4. Immunohistochemical Staining in the Cardiac Myxoma Tissue

The expression and distribution of IL-6 and its receptors, IL-6R and gp130, as well as calretinin in the cardiac myxoma tissue were analyzed by immunohistochemical staining. Cardiac myxoma tissue was fixed with 10% buffered formaldehyde (FUJIFILM Wako Pure Chemical, Osaka, Japan), and embedded in paraffin. Four-μm sections of paraffin-embedded tissue were immunohistochemically stained based on the procedure described in the previous study [53]. The primary antibodies against IL-6, IL-6R, gp130, and calretinin were used at 50-fold dilution. The specificity of the immunostaining was confirmed by substitution of the normal IgG for the primary antibody. Images were taken under microscope at ×400 magnification, and expressions of immunostaining in myxoma were analyzed by microscope (Olympus, Tokyo, Japan).

### 4.5. Immunocytochemical Staining in the Cardiac Myxoma Cells

Myxoma cells plated on a Biocoat slide glass (BD Biosciences, San Jose, CA, USA) were fixed with 1% buffered paraformaldehyde (FUJIFILM Wako Pure Chemical, Osaka, Japan) for 20 min. The indirect immunoperoxidase method was used for the immunocytochemical analysis, as described previously [53]. The primary antibodies against IL-6, IL-6R, gp130, and calretinin were used at a 50-fold dilution. The specificity of the immunostaining was confirmed by substitution of the normal IgG for the primary antibody. Images were taken under microscope at ×400 magnification, and expression of immunostaining in the myxoma cells were analyzed by microscope (Olympus, Tokyo, Japan).

### 4.6. Antibody Array Assay

To identify the protein secreted by the myxoma cells, we used the antibody array of RayBio C-Series Human Cytokine Antibody Array C1000 (AAH-CYT-6) (RayBiotech, Norcross, GA, USA). The culture supernatant of the myxoma cells, incubated for 7 days, was applied to each membrane array, and the expression levels of the proteins were analyzed according to the manufacturer’s instructions.

### 4.7. Western Immunoblot Analysis

The cultured myxoma cells were lysed in an ice-cold cell lysis buffer with a protease inhibitor cocktail. Protein samples resuspended in a sodium dodecyl sulfate buffer and dithiothreitol were separated by 4–12% NuPAGE Bis-Tris gels (Life Technologies, Carlsbad, CA, USA). They were transferred to a polyvinylidene difluoride membrane by electroblotting for 7 min using a Trans-Blot Turbo (Bio-Rad, Hercules, CA, USA). The membrane was incubated with the primary antibody overnight at 4 °C at concentrations recommended by the manufacturer. Subsequently, the membrane was incubated with horseradish peroxidase-conjugated secondary antibody (Cell Signaling Technology, Beverly, MA, USA) for 1 h. The blots were detected using ECL prime (GE Healthcare, Buckinghamshire, UK), and analyzed by a ChemiDoc Touch Imaging System (Bio-Rad, Hercules, CA, USA).

### 4.8. RT-PCR

Total RNA was extracted from the cultured cardiac myxoma cells and HUVECs using a Pure Link RNA Mini kit (Invitrogen, Carlsbad, CA USA). cDNA was synthesized with a Superscript VILO cDNA Synthesis kit (Invitrogen, Carlsbad, CA USA). PCR was performed by using the primer pairs described in Table 1. The specificity of the primers was confirmed by a BLAST search and melting curve analysis. Amplification was performed for 40 cycles with a CFX connect thermal cycler (Bio-Rad, Hercules, CA, USA). The housekeeping gene GAPDH was used as a positive internal control for the PCR action. The PCR products were electrophoretically size-fractionated on an agarose gel, stained with ethidium bromide to visualize DNA bands, and analyzed to determine the presence of the gene.

### 4.9. Real-Time PCR

Semi-quantitative real-time-PCR was performed using Power SYBR Green PCR Master Mix (Applied Biosystems, Warrington, UK) on a CFX connect thermal cycler (Bio-Rad, Hercules, CA, USA). The value of each cDNA was calculated using the ΔΔCq method and normalized to the value of GAPDH.

### 4.10. Transfection with Small Interfering RNA (siRNA)

Transfection with siRNA was performed according to the manufacturer’s protocol (Santa Cruz Biotechnology, Santa Cruz, CA, USA). Transfection complexes were prepared using the siRNA reagent, transfection medium, and STAT3 siRNA, and delivered to cell monolayers with a 100 nmol/L final concentration of siRNA duplexes. A scrambled control siRNA was used as a negative control.

### 4.11. Enzyme-Linked Immunosorbent Assay (ELISA)

Concentrations of IL-6 in the culture medium were determined using a human IL-6 ELISA kit (R&D Systems, Minneapolis, MN, USA) according to the manufacturer’s protocol. Concentrations of IL-6 were determined by comparison of the optical density results with the standard curve.

### 4.12. Immunofluorescence Staining

The cardiac myxoma cells plated on a BioCoat slide glass (BD biosciences, San Jose, CA, USA) were stimulated with IL-6 + sIL-6R in the presence or absence of AG490 for 30 min. The cells were incubated with the antibody against phospho-STAT3 at 50-fold dilution overnight. They were incubated with anti-rabbit IgG-Alexa (Cell Signaling Technology, Beverly, MA, USA) at 250-fold dilution for 1 h, and the nuclei were counterstained with Hoechst 33342 (Invitrogen, Carlsbad, CA, USA) for 5 min. The stained cells were analyzed by fluorescence microscope (Olympus, Tokyo, Japan).

### 4.13. Statistical Analysis

Results of the quantitative studies are expressed as mean ± SEM. Each data point represents the average of three to six independent experiments. A one-way ANOVA test was used to make comparisons among three or more groups and Tukey–Kramer’s post hoc test was used to identify differences between two groups. *p* value < 0.05 was considered statistically significant.

## 5. Conclusions

In conclusion, the present study demonstrated that IL-6 increases its own production and secretion via the activation of STAT3 and Akt pathways in cardiac myxoma cells. Autocrine regulation of IL-6 may play an important role in the pathophysiology of patients with cardiac myxoma.

## Figures and Tables

**Figure 1 ijms-25-02232-f001:**
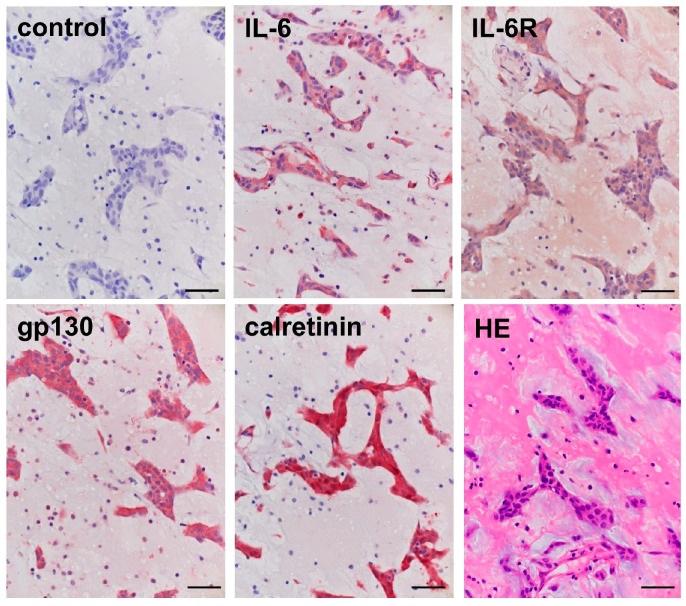
Immunohistochemical staining and hematoxylin-eosin (HE) staining in the cardiac myxoma tissue. Representative immunocytochemical staining showing the localization of IL-6 and its receptor complexes, IL-6R and gp130, and calretinin, together with additional HE staining in the cardiac myxoma tissue. Normal IgG served as a negative control. Original magnification; ×400. Scale bar = 50 μm.

**Figure 2 ijms-25-02232-f002:**
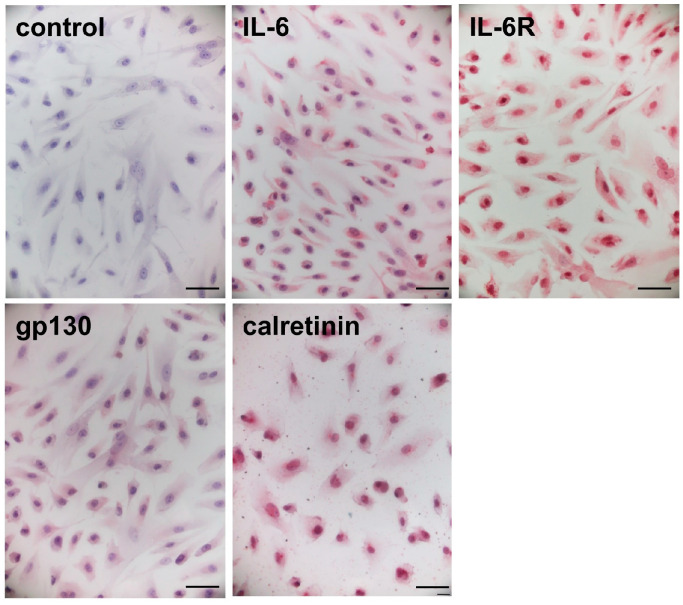
Immunocytochemical staining in the cultured cardiac myxoma cells. Representative immunocytochemical staining showing the localization of IL-6 and its receptor complexes, IL-6R and gp130, and calretinin in the cultured myxoma cells. Normal IgG served as a negative control. Original magnification; ×400. Scale bar = 50 μm.

**Figure 3 ijms-25-02232-f003:**
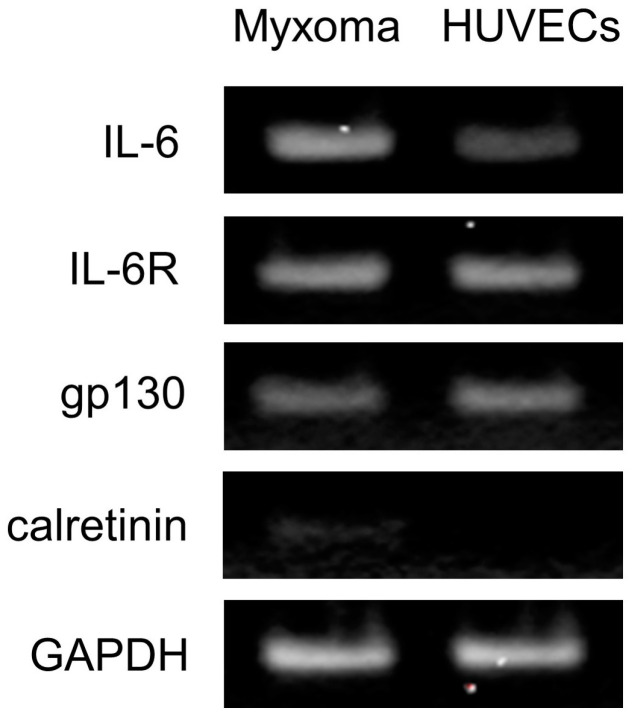
Gene expression of IL-6, IL-6R, gp130, and calretinin in the cardiac myxoma cells. Total RNA was isolated from the cultured cardiac myxoma cells and human umbilical endothelial cells (HUVECs). Reverse transcription-polymerase chain reaction (RT-PCR) was performed using specific primer pairs for IL-6, IL-6R, gp130, calretinin, and glyceraldehyde-3-phosphate dehydrogenase (GAPDH). PCR reaction was performed using 40 cycles and the PCR products of 149, 124, 107, 118 and 138 bp corresponded to the IL-6, IL-6R, gp130, calretinin, and GAPDH transcripts, respectively.

**Figure 4 ijms-25-02232-f004:**
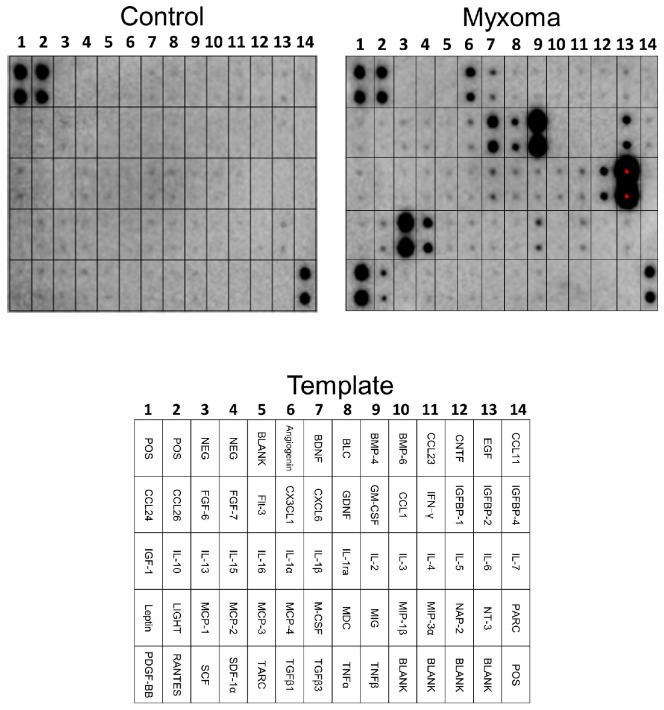
Screening of secreted proteins in the supernatant of the cultured cardiac myxoma cells using antibody array. The culture supernatant of the myxoma cells contained significant amounts of IL-6, monocyte chemoattractant protein (MCP)-1, granulocyte-macrophage colony-stimulating factor (GM-CSF), and platelet-derived growth factor (PDGF)-BB compared with control medium without incubation. POS, positive control; NEG, negative control. For other abbreviations, see RayBio C-Series Human Cytokine Antibody Array C1000 (AAH-CYT-6).

**Figure 5 ijms-25-02232-f005:**
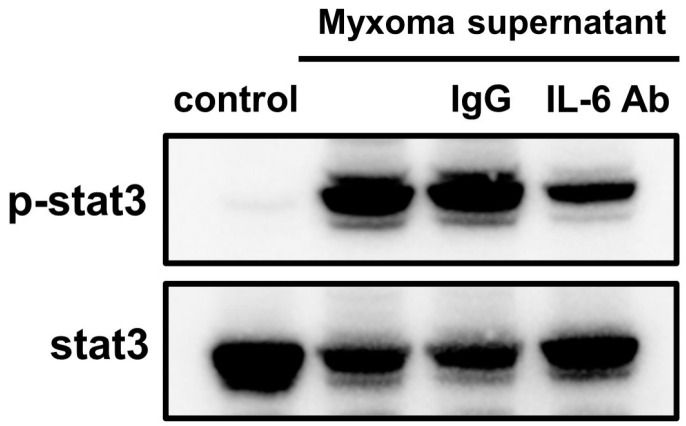
STAT3 phosphorylation induced by the culture supernatant of the cardiac myxoma cells. The culture supernatant derived from the cardiac myxoma cells incubated for 7 days was added to HUVECs and were incubated for 30 min in the presence and absence of anti-IL-6 antibody. Total protein extracts were examined by Western immunoblot analysis.

**Figure 6 ijms-25-02232-f006:**
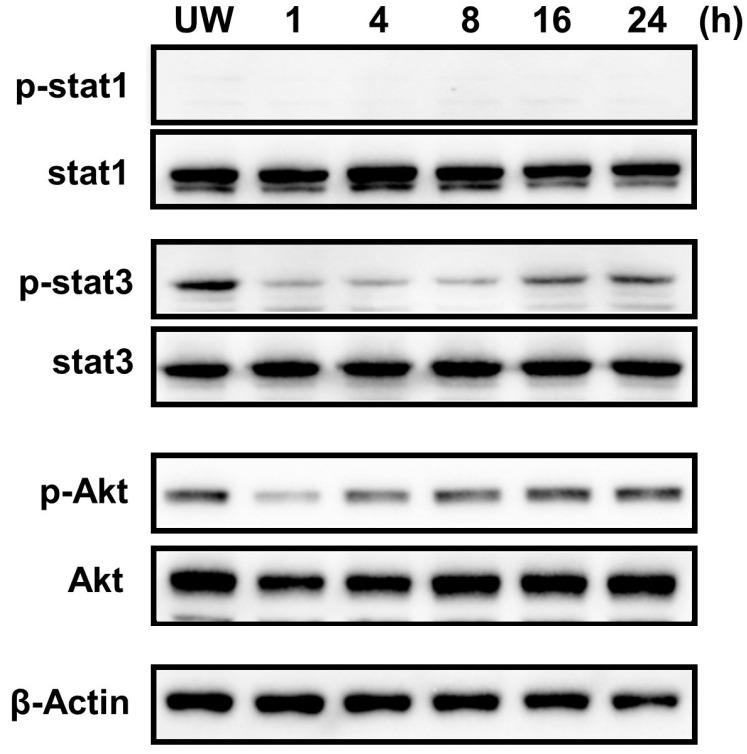
Constitutive phosphorylation of STAT3 and Akt in the cultured cardiac myxoma cells. Cultured cardiac myxoma cells were either unwashed (UW) or washed with phosphate-buffered saline, incubated in fresh culture medium, and left untreated for 1, 4, 8, 16, and 24 h. Total protein extracts were examined by Western immunoblot analysis.

**Figure 7 ijms-25-02232-f007:**
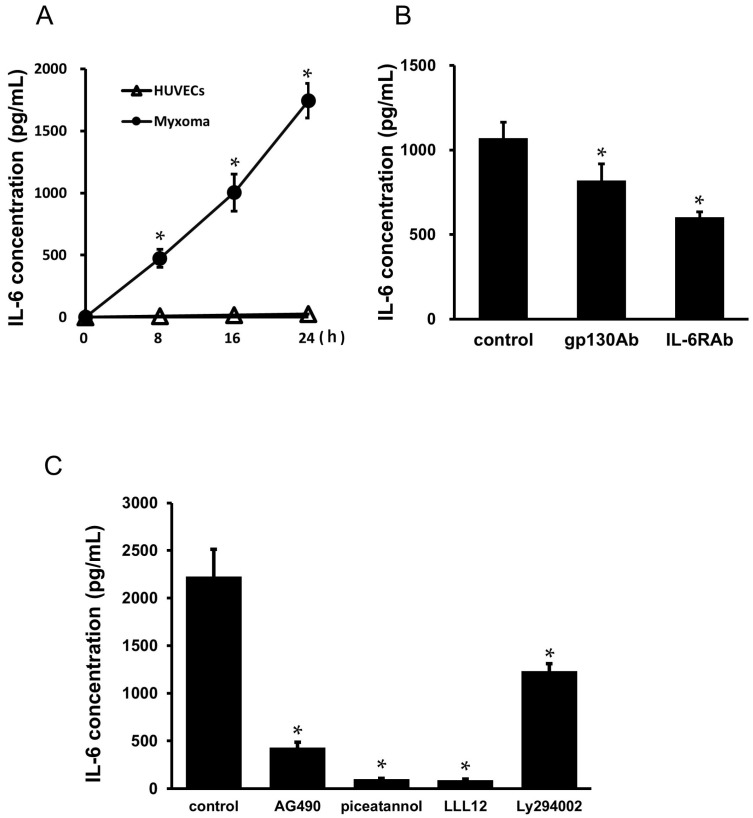
Spontaneous secretion of IL-6 from the cultured cardiac myxoma cells: (**A**) the cultured myxoma cells were washed and incubated in fresh medium. Culture supernatant was collected at 8, 16, and 24 h, measured by ELISA (closed circle), and compared with those of HUVECs (open triangle). IL-6 protein secretion increased in a time-dependent manner. * *p* < 0.05 vs. IL-6 levels in HUVECs at the indicated time; (**B**) the myxoma cells were incubated in fresh medium in the presence or absence of antibody against gp130 or IL-6R. Addition of antibody against gp130 or IL-6R reduced spontaneous secretion of IL-6. * *p* < 0.05 vs. spontaneous IL-6 secretion (control); and (**C**) effects of pharmacological inhibitors of STAT3 and Akt on spontaneous secretion of IL-6 from the cultured myxoma cells. The myxoma cells were pretreated with AG490 (100 μmol/L), piceatannol (100 μmol/L), LLL12 (10 μmol/L), and Ly294002 (30 μmol/L), and IL-6 protein secretion was examined by ELISA (*n* = 6). * *p* < 0.05 vs. IL-6 secretion without any pretreatment (control).

**Figure 8 ijms-25-02232-f008:**
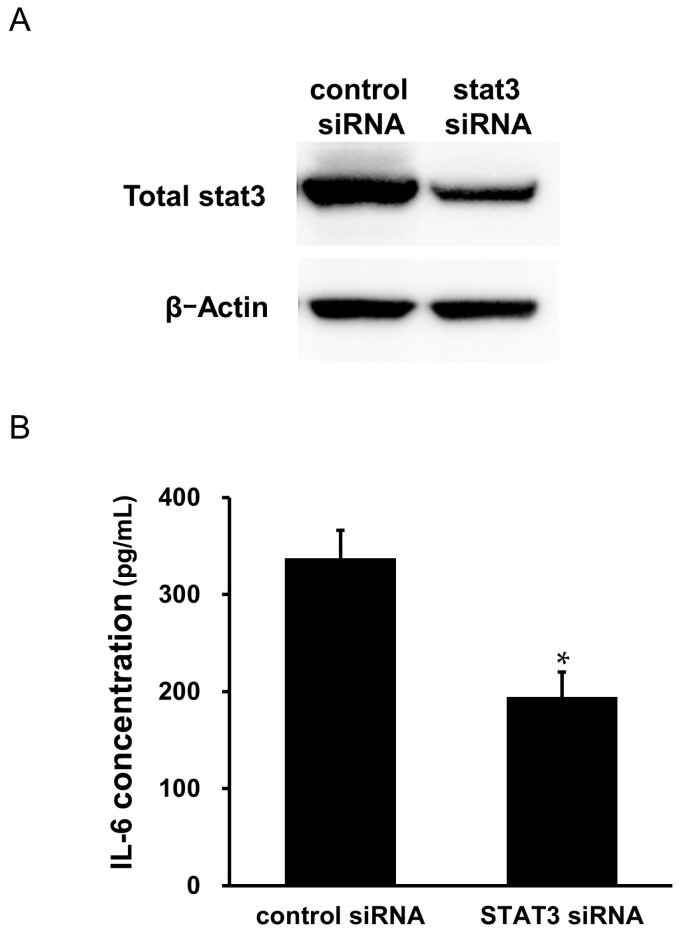
Effects of STAT3 siRNA on the spontaneous secretion of IL-6 from the cardiac myxoma cells. Antibody-free cardiac myxoma cells were transfected with control scrambled siRNA or STAT3 siRNA (100 nmol/L) for 48 h and kept untreated for 24 h: (**A**) cell lysates were evaluated for knock-down of STAT3 by Western immunoblot analysis. β-Actin was used for loading control; and (**B**) IL-6 concentration in the supernatant as measured by ELISA (*n* = 6). Bars represent IL-6 protein secretion per 10^5^ cells (*n* = 6). * *p* < 0.05 vs. scrambled control siRNA.

**Figure 9 ijms-25-02232-f009:**
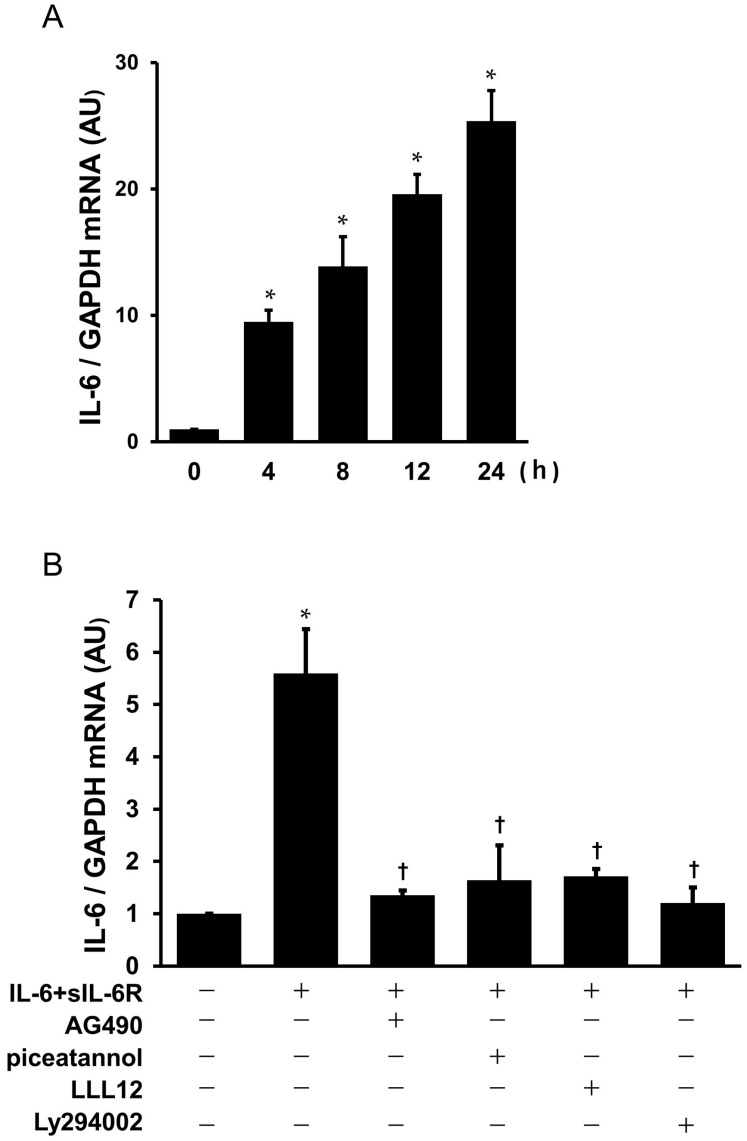
IL-6 + sIL-6R-stimulated IL-6 gene expression in the cardiac myxoma cells: (**A**) time course of IL-6 mRNA after treatment with IL-6 (1 nmol/L) + sIL-6R (1 nmol/L), as evaluated by real-time PCR. Bars represent IL-6 mRNA after normalization to GAPDH mRNA and relative to 0 h (*n* = 3). * *p* < 0.05 vs. 0 h; and (**B**) effects of pharmacological inhibitors of STAT3 and Akt on IL-6 + sIL-6R-stimulated IL-6 mRNA in the cultured myxoma cells. The myxoma cells were preincubated with AG490 (100 μmol/L), piceatannol (100 μmol/L), LLL12 (10 μmol/L), and Ly294002 (30 μmol/L). * *p* < 0.05 vs. the untreated control. † *p* < 0.05 vs. IL-6 + sIL-6R.

**Figure 10 ijms-25-02232-f010:**
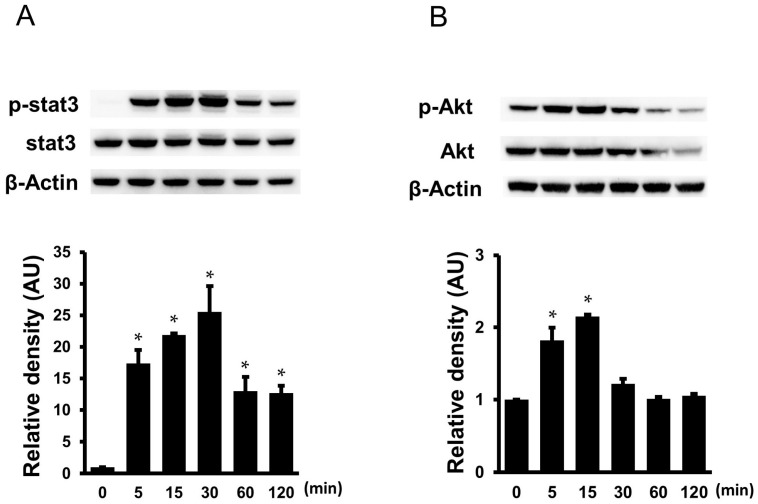
Phosphorylation of STAT3 (**A**); and Akt (**B**) in the cardiac myxoma cells treated with IL-6 (1 nmol/L) + sIL-6R (1 nmol/L). Cultured myxoma cells were treated with IL-6 + sIL-6R for 5, 15, 30, 60, and 120 min. Bars represent densitometric data of each expression signal after normalization to respective total protein and relative to the untreated cells (*n* = 3). * *p* < 0.05 vs. 0 min.

**Figure 11 ijms-25-02232-f011:**
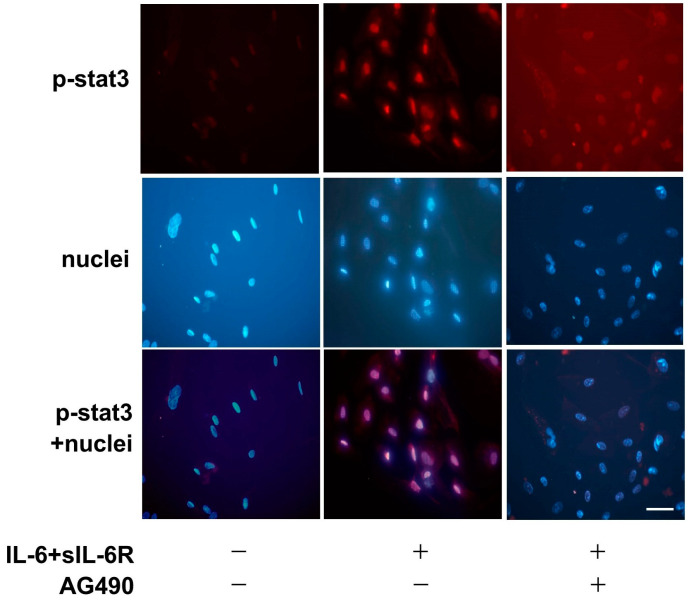
Immunofluorescence staining showing the effects of pharmacological inhibitor of STAT3 on IL-6 + sIL-6R-induced translocation of phosphorylated STAT3 to the nucleus in the cardiac myxoma cells. The cells were pretreated with AG490 (100 μmol/L) followed by additional incubation with IL-6 (1 nmol/L) + sIL-6R (1 nmol/L) for 30 min. Red staining indicates the specific Alexa staining for phosphorylated STAT3, and blue staining indicates the nuclei (Hoechst 33342). Original magnification; ×400. Scale bar = 50 μm.

**Table 1 ijms-25-02232-t001:** Primers of RT-PCR.

Gene Name	Sequence5′-3′	Accession Number
IL-6	(Forward) ACRCACCTCTTCAGAACGAATTG (Reverse) CCATCTTTGGAAGGTTCAGGTTG	NM_000600.3
IL-6R	(Forward) CACGCCTTGGACAGAATCC (Reverse) GCTTGTCGCATTTGCAGAATC	NM_181359
gp130	(Forward) TCAAATCCCTACTCCTTCACTTAC (Reverse) TGGTGAGGAAAATAAACAAGGC	NM_175767
Calretinin	(Forward) TGCCTGTCCAGGAAAACTTC (Reverse) TCATGCTCGTCAATGTAGCC	NM_001740.4
GAPDH	(Forward) GCACCGTCAAGGCTGAGAAC (Reverse) TGGTGAAGACGCCAGTGGA	NM_002046

## Data Availability

The data that support the findings of this study are available from the authors upon reasonable request.

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
