# Peer review of "Autocrine Regulation of Interleukin-6 via the Activation of STAT3 and Akt in Cardiac Myxoma Cells"

_ijms, 2024, doi:10.3390/ijms25042232_

Round 1

Reviewer 1 Report

Comments and Suggestions for Authors

In this report the authors characterize the secretion of IL-6 and IL-6R in vitro by cardiac myoxoma cells.  The authors characterize secretion of IL-6 and IL-6R in vitro, phosphorylation by STAT3 and Akt, the effects of monoclonal blocking of the IL-6R, effects of inhibition of various activation pathways, and gene expression associated with IL-6 and IL-6R.  The manuscript is well written and well organized.

1. I did have one question on data that appeared to be contradictory which was not explained in the discussion.  In Figure 6, the authors show that phorphylation of STAT3 is minimal for the first 24 hours in cultured myxoma cells. In Figure 7A, they demonstrate very rapid increases of IL-6 into the media supernatant by cultured myxoma cells (approx 500 pmol within 8 hours). However, Figure 10 demonstrates that culture of myxoma cells with approximately 1 nmole of IL-6 and IL-6R induces rapid phosphorylation of STAT3 within 5 minutes. Although expression of IL-6R into the supernatant was not reported, Fig 3 demonstrates high levels of IL-6R mRNA expression that appears to be equivalent to IL-6 gene expression and clinical data presented suggested myxoma cells secrete IL-6R.  My question is why Figure 10 demonstrates a rapid STAT3 phosphorylation at concentrations of IL-6 that are comparable to what myxoma cells express in vitro whereas data in Fig 7 indicates STAT3 phosphorylation did not occur within the first 24 hours in vitro.

2. Recognizing that this is outside the scope of the review, presented data in association with the reports on clinical presentations and the suggestion that exogenous IL-6 and IL-6R induce activation pathways on cells other than myxoma cells, made me wonder if normal cardiac myocytes would be activated with possible secretion of cytokines.  If not, what genetic or gene regulatory changes have occurred within myxoma cells that induces secretion of IL-6, IL-6R, GM-CSF, and PAF.

Overall, I found the manuscript to be of high scientific quality and would recommend acceptance.

Reviewer 2 Report

Comments and Suggestions for Authors

The present study investigates IL-6 production via STAT3 and Act pathway in the cardiac myxoma cells and its autocrine regulation playing an important role in the pathophysiology in patients with cardiac myxoma. 

The language seems correct throughout the text and the references are appropriate and up-to-date.

Indeed, this cytokine plays important roles in immune regulation, metabolism and tissue regeneration. Indeed, accumulating evidence has demonstrated that cardiac myxomas constitutively produce considerable amount of IL-6 and that plasma levels of IL-6 positively correlated with tumor size in these patients with cardiac myxoma. However, it would be useful if the authors could report their own opinion, based on current existing literature, of a potential causative role between IL-6 and cardiac myxoma, as also IL-6 association with other molecules produced, suggesting their synergistic role in angiogenesis-related tumor growth, in order to provide aspects of this important reliable biochemical and clinical biomarker improving this interesting clinical model.

Reviewer 3 Report

Comments and Suggestions for Authors

In this study, the authors investigated the role of STAT3 and Akt signal transduction pathways in the autocrine production and secretion of IL-6 in the cardiac myxoma cells. The results showed that IL-6, IL-6R, 97 and gp130 were expressed in the cultured myxoma cells, Levels of IL6, MCP-1, GM-CSF and PDGF-BB in the culture medium in which myxoma cells were cultured for 7 days were increased as compared with those in the control medium. Moreover, treatment with culture supernatant of the myxoma cells induced the STAT3 phosphorylation in HUVECs. Transfection of STAT3 siRNA significantly attenuated the IL-6 secretion from the myxoma cells. It was concluded that demonstrated that IL-6 increases its own production and secretion via the activation of STAT3 and Akt pathways in the cardiac myxoma cells.

Major points:

1. Since HUVECs were used for comparison with myxoma cells in Figure 3, the antibody array study should include the analysis of culture supernatant collected from HUVECs that were cultured for 7 days. 

2. How many independent experiments were performed in the antibody array study?

3. The representative immunofluorescence staining images shown in Figure 11 were not clear and quantitative. The translocation of p-STAT3 to the nucleus should be evaluated via Western blot analysis using the nucleus samples. 

Comments on the Quality of English Language

Minor editing of English language required.
